# Diagnostic performance of a biotin-labeled 4D1 sandwich ELISA for serum antigen detection in talaromycosis

Huamei Wei[1,2], Artid Amsri[1,3], Patcharin Thammasit[1], Kritsada Pruksaphon[4], Yuanji Teng[1,5], Changke Pu[5], Yuefeng Huang[6], Joshua D. Nosanchuk[7], Sirida Youngchim[1]*

1 Department of Microbiology, Faculty of Medicine, Chiang Mai University, Chiang Mai, Thailand, 2 Clinicopathological Diagnosis and Research Center, Affiliated Hospital of Youjiang Medical University for Nationalities, Baise, Guangxi, China, 3 Office of Research Administration, Chiang Mai University, Chiang Mai, Thailand, 4 Department of Medical Technology, School of Allied Health Sciences, Walailak University, Nakhon Si Thammarat, Thailand, 5 Center for Medical Laboratory Science, Affiliated Hospital of Youjiang Medical University for Nationalities, Baise, Guangxi, China, 6 Department of Infectious Diseases, Affiliated Hospital of Youjiang Medical University for Nationalities, Baise, Guangxi, China, 7 Department of Medicine (Division of Infectious Diseases), Department of Microbiology and Immunology, Albert Einstein College of Medicine, New York, United States of America

* syoungchim@gmail.com

## Abstract

### Background

Talaromycosis, caused by the thermally dimorphic fungus *Talaromyces marneffei*, is a life-threatening opportunistic infection in individuals with advanced immunodeficiency, particularly people living with HIV in endemic regions of Southeast Asia. Early diagnosis is essential to reduce morbidity and mortality; however, conventional diagnosis relies mainly on fungal culture, which is time-consuming and has limited sensitivity.

### Methods

We developed and clinically evaluated a direct sandwich enzyme-linked immunosorbent assay (ELISA) using a biotin-labeled monoclonal antibody (4D1) for the detection of *T. marneffei* cytoplasmic yeast antigen in serum. The assay employed the same monoclonal antibody for both capture and detection in a double-recognition format. Analytical performance, cross-reactivity, and clinical diagnostic accuracy were systematically evaluated.

### Results

No cross-reactivity was observed among the tested fungal antigens under the experimental conditions. The analytical limit of detection (LOD) in pooled human serum was 19.398 µg/mL. Clinical evaluation included 79 culture-confirmed talaromycosis cases and 381 non-talaromycosis controls. At an optimized optical density cut-off of 0.268,

**Data availability statement:** All relevant data are within the manuscript and its Supporting information files.

**Funding:** This research was funded by the Overseas Graduate Study Fund from Youjiang Medical University for Nationalities, China, and Thailand Science Research and Innovation (TSRI) for financial support for this work (Grant No. FF 66/016 to HW). This work was also supported by the Guangxi Center for Disease Control and Prevention Science and Technology Project (Grant No. GXJKKJ2025ZC014 to CP, HW). This research was funded by Chiang Mai University (Grant No. TGCMU 2566P009/2566 to SY). The funders had no role in study design, data collection and analysis, decision to publish, or preparation of the manuscript.

**Competing interests:** The authors have declared that no competing interests exist.

the assay demonstrated a sensitivity of 88.61% (95% confidence interval [CI] 79.47-94.66%) and a specificity of 96.06% (95% CI 93.59-97.78%).

## Conclusions

The biotin-labeled 4D1 sandwich ELISA provides a rapid and accurate diagnostic method with good concordance to culture and may support improved clinical diagnosis of talaromycosis. Its potential application for treatment monitoring requires further validation.

## Author summary

Talaromycosis is a severe and potentially fatal fungal infection caused by *Talaromyces marneffei* that predominantly affects individuals with advanced immunodeficiency, especially people living with HIV in endemic regions of Southeast Asia. Delayed diagnosis is strongly associated with increased morbidity and mortality, yet current diagnostic methods rely mainly on fungal culture, which is time-consuming and may lack sensitivity in early disease. To address this limitation, we developed and clinically evaluated a direct sandwich enzyme-linked immunosorbent assay (ELISA) using a biotin-labeled monoclonal antibody (4D1) to detect *T. marneffei* cytoplasmic yeast antigen in serum. The assay uses a double-recognition format with the same antibody for capture and detection, which may improve analytical performance. In clinical testing with culture-confirmed cases and non-talaromycosis controls, the assay demonstrated high sensitivity and specificity, with no cross-reactivity observed among the tested fungal antigens. This diagnostic approach may facilitate earlier detection, support timely antifungal treatment, and potentially enable disease monitoring, thereby improving clinical outcomes in endemic settings.

## 1. Introduction

Talaromycosis is a systemic fungal disease caused by the dimorphic fungus *Talaromyces marneffei* (formerly *Penicillium marneffei*) [1,2]. In 2022, the World Health Organization (WHO) reported an increase in fungal infections among immunocompromised patients, particularly among people living with HIV [3–5]. Epidemiological studies have shown that *T. marneffei* is a major cause of opportunistic infections in AIDS patients, especially in Southeast Asia [6–8]. *T. marneffei* infection is usually disseminated by the time of clinical presentation and involves multiple organs, such as the lungs, liver, skin, and brain. However, its clinical manifestations are often complex and non-specific, which delays diagnosis [9]. Furthermore, studies have shown that early detection and treatment are crucial, as they can reduce the morbidity and mortality rates of the disease [10–12]. To date, the definitive diagnosis of talaromycosis mainly relies on traditional culture methods and morphological identification [9].

These techniques are time-consuming, leading to potential delays in treatment and disease progression [13]. Therefore, there is an urgent need to develop non-culture-based diagnostic methods to improve accuracy, speed, and cost-effectiveness. Such diagnostic approaches may contribute to improved patient outcomes.

Despite considerable advances in molecular and immunological diagnostic techniques for *T. marneffei*, several limitations persist among existing methods [14–23]. Metagenomic next-generation sequencing (mNGS), although highly effective, remains difficult to implement widely in routine clinical practice because of its high cost and limited accessibility [24]. Nested PCR can improve both sensitivity and specificity for *T. marneffei* detection. However, it requires two sets of primers (external and internal), involves a relatively complex workflow, and is time-consuming [25]. On the other hand, immunological analysis may produce different results in different studies with the same target, such as the detection of galactomannan (GM) and mannoprotein 1 (Mp1p) [16,26,27]. The inconsistent GM results may be attributed to the prevalence of GM in the fungal cell wall, which is prone to cross-reactivity. Mp1p is an immunogenic surface and secreted mannoprotein of *T. marneffei* [28], which is expected to be a diagnostic target. One study reported high sensitivity (97.90%) and specificity (100%) by using a combination of monoclonal and polyclonal antibodies to simultaneously detect Mp1p antigen and antibody in serum samples [20]. However, rare false-positive results have been observed in some clinical settings, suggesting that nonspecific binding or untested fungal species may still contribute to occasional cross-reactions [22], thereby potentially limiting the assay's specificity. Therefore, the development of a diagnostic method for *T. marneffei* infection that combines high sensitivity and specificity, cost-effectiveness, and operational simplicity remains essential in clinical practice [5].

Monoclonal antibody (mAb) 4D1 is a highly specific immunoglobulin for detecting a cytoplasmic yeast-phase antigen of *T. marneffei* [29]. Previous studies from our group have demonstrated the diagnostic potential of mAb 4D1 in immunoassays for talaromycosis. Recently, we developed a sandwich ELISA combining mAb 4D1 with *Galanthus nivalis* agglutinin (GNA) [27], a mannose-binding lectin capable of recognizing cytoplasmic yeast antigens of *T. marneffei* [30–32]. Subsequently, we developed an innovative immunochromatographic test (ICT) employing mAb 4D1 for the detection of *T. marneffei* antigens in urine samples [33]. The assay yielded significant results within 20 minutes in urine samples containing low antigen concentrations (3.125 µg/mL) and demonstrated excellent diagnostic performance, with 87.87% sensitivity, 100% specificity, and 95.5% overall accuracy. The high specificity and unique recognition properties of mAb 4D1 highlight its diagnostic relevance for *T. marneffei* infection [27,30]. Moreover, the antibody reacts with a highly glycosylated, high-molecular-weight mannoprotein ranging from 50 to 150 kDa, providing a molecular basis for the development of sensitive and specific diagnostic assays [34]. Importantly, mAb 4D1 exhibits no cross-reactivity with the tested fungal antigens [35].

Despite these advantages, urine-based assays may exhibit reduced sensitivity in certain patients, as *T. marneffei* antigens may be absent or present at undetectable levels in urine during early infection or due to limitations in glomerular filtration [36]. Moreover, studies on other invasive fungal infections have demonstrated that serum can serve as a reliable specimen type for molecular detection, often showing improved performance compared with whole blood [37,38]; consistent with these observations, serum has also been suggested to be a practical and reliable specimen type for the detection of *T. marneffei* [39,40]. However, our research group found that the GNA-mAb 4D1 ICT strip was ineffective when applied to serum samples, owing to strong nonspecific interactions between serum components and GNA, which interfered with antigen detection [30]. Therefore, these findings indicated the need to consider alternative diagnostic strategies or detection systems capable of overcoming serum matrix interference, thereby enabling more accurate and sensitive diagnosis of *T. marneffei* infection.

Biotin (vitamin B7 or vitamin H; cis-hexahydro-2-oxo-1H-thieno [3,4-d] imidazole-4-pentanoic acid) offers unique advantages for immunoassay development due to its small molecular weight (240 Da) and flexible structure. These properties enable efficient side chain labeling of proteins without compromising their biological activity [41–44]. Structurally, biotin consists of two fused heterocyclic rings (Fig A in S1 File). One ring contains a ureido group that mediates

strong non-covalent binding to avidin, while the second ring comprises a tetrahydrothiophene moiety linked to a valeric acid side chain. The carboxyl group of this side chain enables covalent conjugation of biotin to a wide range of proteins, thereby preserving functional integrity while supporting downstream biological and analytical applications [45]. Consistent with these properties, Nguyen and colleagues employed biotin-conjugated Mp1p mAbs for detecting *T. marneffei* antigens, achieving sensitivities and specificities of 86.3% and 98.1%, respectively [22]. Similarly, studies on other fungal pathogens have demonstrated that dual-mAb sandwich ELISAs could achieve high diagnostic accuracy without cross-reactivity [46].

Given these challenges, this study aimed to enhance the diagnostic performance of *T. marneffei* detection by optimizing an existing mAb 4D1-based immunoassay. We developed a direct sandwich ELISA employing mAb 4D1 as the capture antibody and a biotin-labeled 4D1 as the detector antibody for the identification of cytoplasmic yeast antigens of *T. marneffei* in serum samples. Incorporation of the biotin-labeled 4D1 enabled signal amplification and may enhance antigen detection, thereby improve assay sensitivity while maintain high specificity and no cross-reactivity with the tested fungal antigens in this study. This optimized biotin-labeled 4D1 sandwich ELISA represents a potentially useful diagnostic tool for talaromycosis. Our data suggest that the biotin-labeled 4D1 sandwich ELISA is a promising approach for early disease detection, therapeutic monitoring, and assessment of antifungal treatment response, thereby potentially supporting improved clinical management in endemic settings.

## 2. Materials and methods

### Ethics statement

This study was approved by the Research Ethics Committee of the Faculty of Medicine, Chiang Mai University (No. EC0090/2568), and the Research Ethics Committee of the Affiliated Hospital of Youjiang Medical University for Nationalities (No. ECYYFY-2025–279). Written informed consent was obtained from all participants prior to enrollment in the study.

### 2.1 Serum samples

All samples were collected between September 2023 and October 2025 from the Clinical Pathology Unit, Maharaj Nakorn Chiang Mai Hospital (a tertiary hospital affiliated with Chiang Mai University), Chiang Mai, Thailand, and the Department of Clinical Laboratory, Affiliated Hospital of Youjiang Medical University for Nationalities (Fig 1). The samples were stored at -80 °C until use.

The diagnostic performance of the biotin-labeled 4D1 sandwich ELISA was evaluated using 460 clinical serum samples (79 patients with a positive *T. marneffei* culture were enrolled as the positive group, 309 patients with negative *T. marneffei* cultures, and 72 healthy individuals were enrolled as the negative control group) (Table 1). The non-*T. marneffei* control group consisted of 381 participants, including 139 patients with other microbial infections (Table B in S1 File), 170 patients without evidence of infection, and 72 healthy individuals from endemic areas. The 309 patients with or without additional isolated pathogens were epidemiologically similar to the 79 patients with *T. marneffei*. Positive *T. marneffei* cultures were confirmed by thermal dimorphic transition between 25 °C and 37 °C and microscopic analysis for phenotypic characterization. The cohort included both male (53) and female (26) patients, aged from 1 to 74 years (Table A in S1 File). The sample size was estimated based on the expected diagnostic performance of the sandwich ELISA. The parameters were set as follows: a type I error rate (α) of 0.05, a 95% confidence level, and a desired precision (margin of error) of 10%. The expected sensitivity and specificity were both set at 0.9, based on previously reported performance of similar immunoassays for *T. marneffei* [39,40]. Sample size estimation was performed according to the methods described by Buderer and Hajian-Tilaki [47,48]. Although no universal minimum sample size exists for diagnostic accuracy studies, previous work has suggested that studies with several dozen cases may provide initial evidence under certain conditions [49–51]. Accordingly, a minimum sample size of approximately 70 participants was considered sufficient for this study. Further

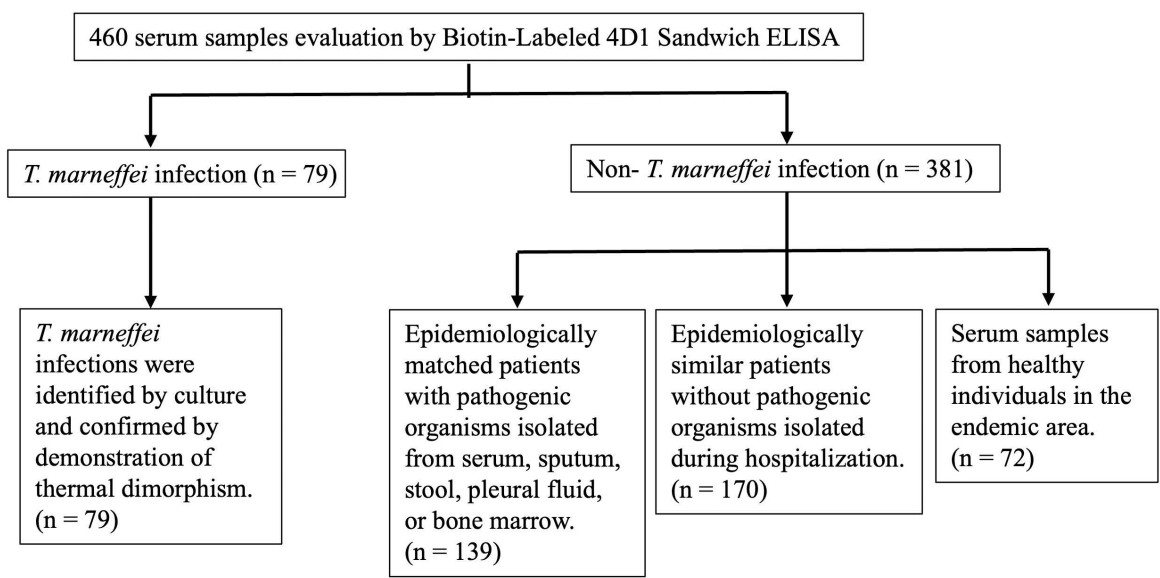

**Fig 1. Flow diagram of the selection of 460 clinical samples included in this study.** The culture-confirmed *T. marneffei* infection group included 79 patients (Table A in S1 File). The non-*T. marneffei* control group consisted of 381 participants, including 139 patients with other microbial infections (Table B in S1 File), 170 patients without evidence of infection, and 72 healthy individuals from endemic areas. The 309 patients with or without additional isolated pathogens were epidemiologically similar to the 79 patients with *T. marneffei*. The serum samples were obtained from the Clinical Pathology Unit, Maharaj Nakorn Chiang Mai Hospital (a tertiary hospital affiliated with Chiang Mai University), Chiang Mai, Thailand, and the Department of Clinical Laboratory, Affiliated Hospital of Youjiang Medical University for Nationalities, China.

**Table 1. Baseline clinical characteristics of study participants.**

| Group | Total (n) | Male, n (%) | Female, n (%) | Age (mean±SD, years) | CD4+T-cell count (median, IQR; cells/mm³) |
|---|---|---|---|---|---|
| *T. marneffei* infection | 79 | 53 (67.1) | 26 (32.9) | 40.0±15.2 | 15.0 (5.0-32.0) |
| *T. marneffei*-negative, other microbial infections | 139 | 81 (58.3) | 58 (41.7) | 49.5±19.6 | 122.0 (39.5-390.5) |
| *T. marneffei*-negative, no microbial infection | 170 | 104 (61.2) | 66 (38.8) | 50.9±18.3 | 186.0 (37.0-439.0) |
| Endemic-area healthy population | 72 | NA | NA | NA | NA |

**Notes:** NA, not available. Data are presented as mean±standard deviation (SD) or median with interquartile range (IQR), as appropriate. Sex percentages were calculated based on available data. CD4+T-cell counts were available for 43 of 79 patients with *T. marneffei* infection, 50 of 139 *T. marneffei*-negative patients with other microbial infections, and 73 of 170 *T. marneffei*-negative patients without microbial infection. Missing CD4+T-cell data were due to incomplete laboratory testing at the time of serum collection. For the endemic-area healthy population, no individual-level clinical metadata (including sex, age, or CD4+T-cell counts) were available.

details regarding samples from patients with infections caused by other pathogenic microorganisms are available in the supplementary materials (Table B in S1 File).

## 2.2 *T. marneffei* cytoplasmic yeast antigen (TMCYA) preparation

Briefly, *T. marneffei* conidia were inoculated into brain heart infusion (BHI) broth (Becton Dickinson, Sparks, MD, USA) and cultured for 6 days at 37 °C, 150 rpm. After treatment with 0.02% (*w/v*) Merthiolate (Sigma-Aldrich, Poole, UK) at room

temperature for 12 h, the yeasts were centrifuged for 10 min at 2,300 × g and the cell walls were mechanically broken with 0.5-mm glass beads (BioSpec, Bartlesville, OK, USA) in a bead beater homogenizer (BioSpec, Bartlesville, OK, USA). A cocktail of protease inhibitors including iodoacetic acid (IAA; 1 mM; GE Healthcare, Little Chalfont, Buckinghamshire, UK), phenylmethanesulfonyl fluoride (PMSF; 0.1 mM; Tokyo Chemical Industry Co., Ltd., Tokyo, Japan), and 1 mM ethylene diamine tetraacetic acid (EDTA; Merck, Darmstadt, Germany), were added to prevent fungal protein degradation. The mixture was centrifuged for 30 min at 4 °C at 10,000 × g, and the total antigen was collected [52]. The protein concentration of the antigen preparations was determined using a bicinchoninic acid (BCA) protein assay kit (Thermo Fisher Scientific, USA) according to the manufacturer's instructions. The antigens of other fungi were prepared following the same standardized procedure. The fungal strains used in the current study are listed in Table 2.

## 2.3 Purification of mAb 4D1

The murine-derived hybridoma cell line 4D1 was cultured in hybridoma serum-free medium (Gibco, Grand Island, NY, USA). The culture supernatant of mAb 4D1 was concentrated with a Vivaspin 20 ultrafiltration device (MWCO: 30 kDa; GE Healthcare, Little Chalfont, Buckinghamshire, UK). The supernatant was filtered through a 0.45 μm membrane filter (Corning Incorporated, Corning, NY, USA). After that, the concentrated 4D1 supernatants were purified by ÄKTA start chromatography system (Cytiva, Uppsala, Sweden) with HiTrap protein G affinity chromatographic column (GE Healthcare, Uppsala, Sweden). Protein G was covalently linked to Sepharose bead for capturing mAb 4D1, which belongs to the mouse IgG1 subclass. MAb 4D1 was eluted by 0.1 M glycine-HCl pH 2.7. Then, the purified antibody solution was immediately neutralized by adding 60 μL of 1 M Tris-HCl pH 9.0. Eluted fractions were screened by indirect ELISA. The eluted fractions containing mAb 4D1 were pooled and dialyzed in PBS pH 7.4. Final mAb 4D1 concentrations were measured by Pierce BCA Protein Assay Kit (Thermo Fisher Scientific, Rockford, IL, USA).

The purity of mAb 4D1 was validated using 10% SDS-PAGE. To confirm the mouse IgG subclass of mAb 4D1 by immunoblotting, SDS-PAGE gels containing separated polypeptides were transferred electrophoretically to nitrocellulose membranes (Hybond extra; Amersham, Little Chalfont, Buckinghamshire, UK). The membranes were washed with PBS and then blocked overnight at 4 °C with PBS-0.05% Tween 20 containing 5% skim milk (Sigma-Aldrich, St. Louis, MO,

**Table 2. Antigen preparations were derived from the diverse fungal isolates.**

| Fungal species | Isolate number |
|---|---|
| *Talaromyces marneffei* | ATCC 200051[#] |
| *Cryptococcus neoformans* | H99[#] |
| *Candida albicans* | ATCC 900028[#] |
| *Sporothrix schenckii* | 52-S1[*] |
| *Aspergillus fumigatus* | 55-A1[*] |
| *Penicillium citrinum* | MMC59P12-1[$] |
| *Scedosporium apiospermum* | MMC60P21-1[$] |
| *Geotrichum spp.* | CI |
| *Pichia kudriazevii* (*Candida krusei*) | CI |
| *Trichosporon spp.* | CI |

#: Isolate from American Type Culture Collection, Rockville, MD, USA;

*: Isolate from the Institute of Dermatology, Department of Medical Services, Ministry of Public Health, Bangkok, Thailand;

$: Isolate from culture collection in Mycology Unit, Department of Microbiology, Faculty of Medicine, Chiang Mai University, Chiang Mai, Thailand;

CI: Clinical isolates from blood samples of infected patients.

USA). Later, the membranes were washed with PBS-0.05% Tween 20 and incubated for 1 hour with HRP-conjugated goat anti-mouse IgG (Jackson, West Grove, PA, USA) diluted 1:3000 in PBS-0.05% Tween 20. The membranes were washed, and the bound conjugates were visualized by incubation with TMB/$H_2O_2$ chromogenic substrate (BioFX Laboratories, Surmodics, Eden Prairie, MN, USA). The reactions were stopped by submersion of the membrane in distilled water.

### 2.4 Biotin-4D1 IgG conjugation

The conjugation of mAb 4D1 IgG with biotin was performed according to the manufacturer's (Thermo Fisher Scientific, Waltham, MA, USA) instructions. Briefly, the antibody was buffer-exchanged to remove amine-containing components by dialysis against 0.1 M sodium phosphate, 0.15 M sodium chloride (pH 7.2) for 1 week. The dialyzed antibody solution was concentrated using 10 kDa Vivaspin 6 centrifugal concentrators, and the protein concentration determined by BCA assay. Then, 2 mg/mL of purified 4D1 IgG was reacted with freshly prepared 10 mM Sulfo-NHS-LC-Biotin (Thermo Fisher Scientific, Waltham, MA, USA) in ultrapure water. For conjugation, 27 µL of biotin solution was added per 1 mL of antibody solution and incubated on ice for 2 hours. Following conjugation, 10 mM Tris-HCl was added and incubated for 1 hour to quench unreacted NHS esters. The biotinylated antibody was then dialyzed extensively for 1 day to remove unbound biotin before storage.

### 2.5 Verification of biotin-labeled 4D1 conjugation

Validation of the biotin-labeled 4D1 conjugation was performed using 10% SDS-PAGE and Western blotting. A total of 12 protein antigens from 10 fungal species were analyzed by SDS-PAGE, including cytoplasmic antigens of *T. marneffei* obtained from its conidial, mold, and yeast phases. The same set of proteins was loaded onto five identical gels at a final concentration of 10 µg per lane. After electrophoresis, the separated polypeptides were transferred onto nitrocellulose membranes. One gel was stained with Coomassie Brilliant Blue to visualize protein bands, whereas the remaining four gels were used for membrane transfer.

Following transfer, the membranes were blocked overnight at 4 °C in 3% (*w/v*) BSA in 20 mM Tris-HCl (pH 7.4) containing 0.15 M NaCl and 0.05% Tween-20. After blocking, all membranes were washed with PBS-0.05% Tween-20. The first membrane was incubated with mAb 4D1 for 2 hours, followed by incubation with HRP-conjugated goat anti-mouse IgG (Jackson ImmunoResearch, West Grove, PA, USA) diluted 1:3000 in PBS-0.05% Tween-20 for 1 hour. The second membrane was incubated with biotin-labeled 4D1 (20:1, diluted 1:1000 in 1% (*w/v*) BSA prepared in 10 mM Tris-HCl, pH 7.4) for 2 hours and subsequently incubated with streptavidin-HRP diluted 1:6000 in 1% (*w/v*) BSA prepared in 10 mM Tris-HCl (pH 7.4) for 1 hour. The remaining two membranes served as negative controls. For these controls, mAb 4D1 and biotin-labeled 4D1 were omitted and replaced with either PBS-0.05% Tween-20 or 1% (*w/v*) BSA in 10 mM Tris-HCl (pH 7.4), respectively, prior to incubation with HRP-conjugated goat anti-mouse IgG (1:3000) or streptavidin-HRP (1:6000) for 1 hour. After the final washing steps, bound conjugates were visualized using a TMB/$H_2O_2$ chromogenic substrate. The reactions were stopped by rinsing the membranes in distilled water.

### 2.6 Biotin-labeled 4D1-based sandwich ELISA procedure

Fig 2 illustrates the principle of the biotin-labeled 4D1-based sandwich ELISA. Briefly, 100 µL of purified 4D1 monoclonal antibody (50 µg/mL in 0.06 M carbonate buffer, pH 9.6) was coated onto MaxiSorp 96-well plates (Greiner Bio-One GmbH, Frickenhausen, Germany) and incubated at 37 °C for 90 minutes. The wells were then washed six times with phosphate-buffered saline containing 0.1% Tween-20 (PBST) and blocked with 200 µL of 3% (*w/v*) BSA in 20 mM Tris-HCl (pH 7.4) containing 0.15 M NaCl and 0.05% Tween-20 for 2 hours at room temperature. After blocking, 100 µL of serum samples (diluted 1:10 in PBS containing 0.1% BSA and 0.05% Tween-20) was added to each well in triplicate and incubated for 2 hours at room temperature.

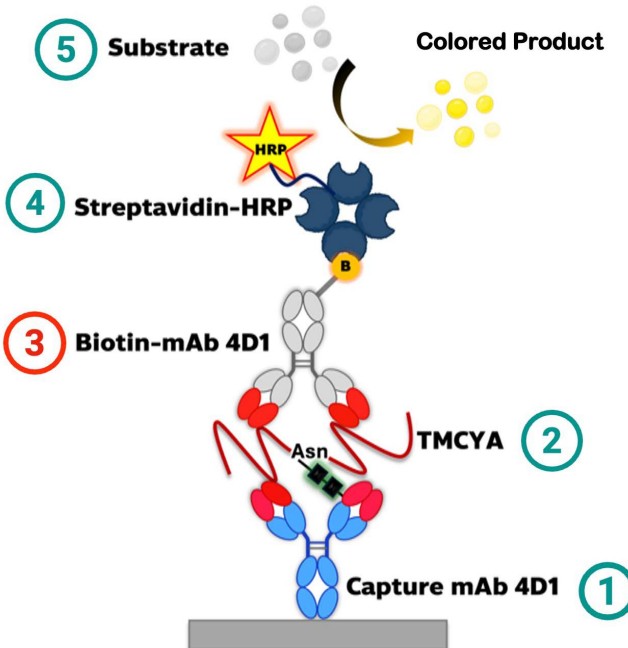

**Fig 2. Schematic representation of the biotin-labeled 4D1-based sandwich ELISA for the detection of TMCYA. (1)** Capture mAb 4D1 is coated on the solid phase. **(2)** TMCYA antigen binds to the immobilized antibody via its Asn(Asparagine)-containing epitope. **(3)** Biotin-labeled 4D1 detection antibody binds to a second epitope on TMCYA, forming the sandwich complex. **(4)** Streptavidin-HRP associates with the biotin-labeled 4D1. **(5)** Addition of chromogenic substrate yields an HRP-catalyzed colored product for detection.

Following washing, 100 µL of biotin-labeled 4D1 antibody (1:1000 dilution in 10 mM Tris-HCI containing 3% BSA) was added and incubated for 90 minutes. Then, the wells were washed and incubated with 100 µL of streptavidin-HRP (1:6000 dilution in 10 mM Tris-HCI with 3% BSA) for 1 hour at room temperature.

After a final wash step, 100 µL of 3,3',5,5'-tetramethylbenzidine (TMB; Surmodics IVD, Eden Prairie, MN, USA) substrate was added and allowed to develop for 15 minutes before stopping the reaction with 50 µL of 2 M $H_2SO_4$. Optical density was measured at 450 nm (ELISA reader, Shimadzu model UV-2401PC, Kyoto, Japan).

Key assay parameters, including coating antibody concentration, serum dilution, biotin-labeled 4D1 dilution, and streptavidin-HRP dilution, were systematically optimized through preliminary serial dilution experiments. For the optimization experiments, TMCYA-spiked serum samples were prepared at an initial concentration of 100 µg/mL. Serial dilutions were performed stepwise using negative serum or appropriate buffer to obtain dilution ratios ranging from 1:2–1:100. Each subsequent dilution was prepared from the preceding solution. The performance of each condition was evaluated based on both the P/N ratio and the difference between the mean OD values of positive and negative samples (ΔOD), allowing identification of optimal assay parameters with maximal signal discrimination and minimal background interference. The selected conditions (5 µg/well coating antibody, 1:1000 biotin-labeled 4D1 (20:1), 1:6000 streptavidin-HRP, and 1:10 serum dilution) were determined based on these optimization experiments. The corresponding optimization results are presented in the Supplementary Materials (Tables C-E and Figs D-F in S1 File).

## 2.7 Application of the ELISA: standard curve, specificity, and sample testing

To evaluate the performance of the biotin-labeled 4D1 sandwich ELISA, serum samples were collected from ten healthy volunteers residing in *T. marneffei*-endemic areas. The sera were pooled to generate a reference sample. This pooled

serum was used for two purposes: (1) preparation of a standard curve via serial dilution to establish assay linearity and dynamic range, and (2) evaluation of assay specificity by comparing the biotin-labeled 4D1 sandwich ELISA signal with clinical sera from patients with culture-confirmed *T. marneffei* infection. All samples were derived from the same cohort described in Section 2.1 and were collected under the same institutional ethics approvals.

**2.7.1 Standard curve construction.** For standard curve construction, pooled serum samples were diluted 1:10 and spiked with TMCYA at concentrations ranging from 0.781-1000 µg/mL. The biotin-labeled 4D1 sandwich ELISA was performed as described in Section 2.6 to generate a standard curve and assess assay linearity and dynamic range.

The standard curve was fitted using a four-parameter logistic (4-PL) regression model of the form:

$$Y = D + (A - D) / [1 + (X/C)^B],$$

where A and D represent the asymptotic minimum and maximum responses, B is the slope factor, and C is the inflection point (EC50) [53,54]. This model was used to interpolate antigen concentrations from OD values.

**2.7.2 Specificity testing.** To assess assay specificity, pooled serum samples were spiked with TMCYA or antigens from other fungal species at the same concentration range (0.781-1000 µg/mL). The assay was performed as described in Section 2.6, and the resulting signals were compared to evaluate potential cross-reactivity.

**2.7.3 Sample testing.** Serum samples from patients with culture-confirmed *T. marneffei* infection and control subjects were analyzed using the same assay procedure. The optical density values were interpreted using the established standard curve to determine the sensitivity, specificity, and overall diagnostic performance of the assay.

**2.7.4 Determination of LOB, LOD, and LOQ.** The analytical performance of the biotin-labeled 4D1 sandwich ELISA was evaluated by determining the limit of blank (LOB), limit of detection (LOD), and limit of quantification (LOQ) [55]. Blank samples consisted of pooled serum without added antigen. The LOB was calculated as the mean optical density (OD) of blank samples plus $1.645 \times$ standard deviation (SD). The LOD was defined as the LOB plus $1.645 \times$ SD of low-concentration samples [55]. The LOQ was defined as the lowest antigen concentration that could be quantified with acceptable precision and accuracy, corresponding to a coefficient of variation (CV) of less than 20% [56], as commonly applied in analytical assay validation. The OD values corresponding to LOD and LOQ were converted to antigen concentrations using the four-parameter logistic (4-PL) standard curve.

## 2.8 Statistical analysis

Statistical analyses and graphing were performed using GraphPad Prism (version 10.0; GraphPad Software, San Diego, CA, USA) and MedCalc Statistical Software (version 23.4; MedCalc Software Ltd., Ostend, Belgium). GraphPad Prism was used to generate four-parameter logistic (4-PL) standard curves, receiver operating characteristic (ROC) curves, and to calculate the area under the curve (AUC), limit of detection (LOD), and optimal cut-off values. Diagnostic accuracy parameters, including Cohen's kappa (κ), sensitivity, specificity, positive predictive value (PPV), negative predictive value (NPV), and their 95% confidence intervals (CIs), were calculated using MedCalc. Positive (LR+) and negative likelihood ratios (LR-) were derived to evaluate the diagnostic performance. $P < 0.05$ was considered statistically significant.

## 3. Results

### 3.1 Purification of mAb 4D1

The purity of the mAb 4D1 fractions was assessed at the elution step by monitoring the UV absorbance peak (Fig BA in S1 File). Subsequently, the collected fractions were analyzed by SDS-PAGE (Fig BB in S1 File). The IgG subclass of mAb 4D1 was determined by Western blot with HRP-conjugated goat anti-mouse IgG, confirming that mAb 4D1 belongs to the IgG1 subclass (Fig BC in S1 File).

## 3.2  Verification of biotin-4D1 conjugation

Successful conjugation of biotin-labeled 4D1 was confirmed by SDS-PAGE and Western blotting (Fig 3). In the SDS-PAGE gel stained with Coomassie Brilliant Blue, all ten fungal cytoplasmic antigen preparations were clearly resolved, displaying a broad molecular weight distribution (Fig 3A). When the membrane was probed with the mAb 4D1 followed by HRP-conjugated goat anti-mouse IgG, a distinct immunoreactive signal was observed exclusively for TMCYA, which migrated between approximately 50–150 kDa, whereas no detectable reactivity was observed for other fungal antigens (Fig CA in S1 File). This result is consistent with the findings reported by Pruksaphon et al. [34]. Similarly, incubation with the biotin-labeled 4D1 followed by streptavidin-HRP resulted in a detectable signal exclusively for TMCYA, whereas all other fungal antigens remained non-reactive (Fig 3B). As negative controls, membranes incubated with HRP-conjugated goat anti-mouse IgG or with streptavidin-HRP (without addition of mAb 4D1 or biotin-labeled 4D1) showed no detectable signal for any of the fungal antigens (Fig CB and CC in S1 File). Collectively, these results confirm successful biotin labeling of mAb 4D1 and demonstrate that the conjugate retains high specificity for TMCYA, supporting its use in subsequent experiments.

## 3.3  Standardization of the biotin-labeled 4D1 sandwich ELISA

A standard curve for the biotin-labeled 4D1 sandwich ELISA was established using pooled serum spiked with TMCYA at concentrations ranging from 0.781 to 1000 µg/mL (Fig 4). Serial dilutions were assayed with two replicate wells per concentration, and the mean $OD_{450}$ values from three independent experiments were used to generate the curve. The four-parameter logistic (4-PL) regression demonstrated excellent reproducibility and a strong concentration-dependent response. The coefficient of determination ($R^2$) for the fitted curve was 0.997, indicating that 99.7% of the variability in $OD_{450}$ values could be explained by the TMCYA concentration. This high $R^2$ value reflects a highly reliable model with

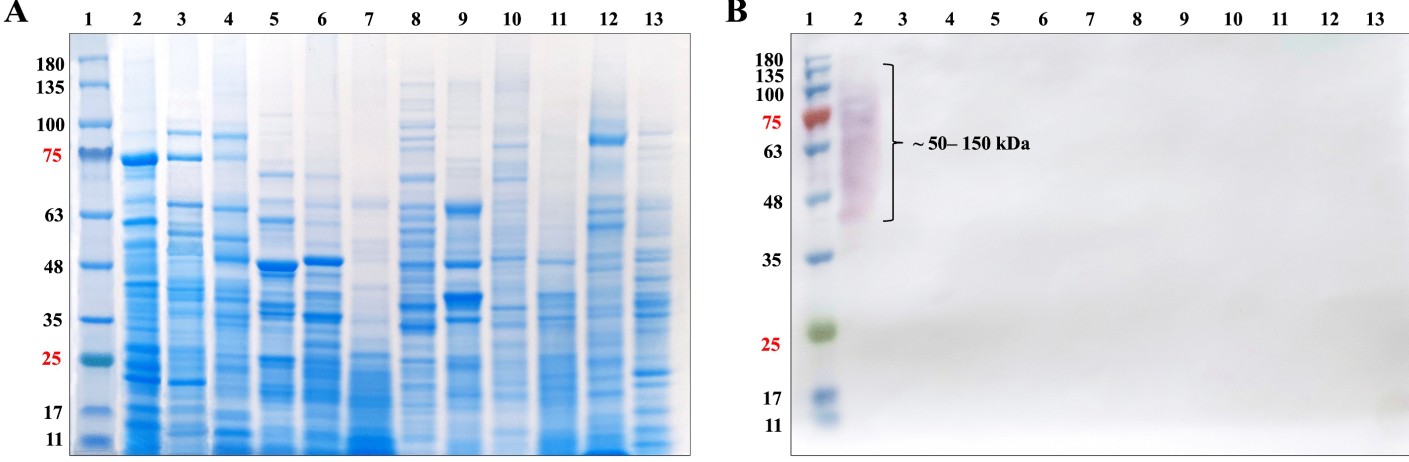

**Fig 3.  Verification of biotin-labeled 4D1 antibody conjugation and its binding specificity for TMCYA.** (A) Coomassie Brilliant Blue-stained 10% SDS-PAGE of ten fungal cytoplasmic antigen preparations (10 µg/well), demonstrating successful protein extraction and clear band separation across a broad molecular weight range (11-180 kDa). (B) Western blot of the same antigen panel probed with biotin-labeled 4D1 followed by streptavidin-HRP detection. A positive signal was observed exclusively in the TMCYA lane (lane 2), producing a broad immunoreactive band spanning approximately 50-150 kDa. No reactivity was detected against cytoplasmic antigens derived from *T. marneffei* conidia or mold forms, or from other fungal species tested under the experimental conditions, supporting the high specificity of the 4D1 conjugate for TMCYA. Lane assignments are as follows: lane 1, molecular weight marker; lane 2, *T. marneffei* yeast; lane 3, *T. marneffei* conidia; lane 4, *T. marneffei* mold; lane 5, *C. albicans*; lane 6, *P. kudriazevii*; lane 7, *Geotrichum sp.*; lane 8, *S. apiospermum*; lane 9, *Trichosporon sp.*; lane 10, *S. schenckii* (yeast); lane 11, *C. neoformans*; lane 12, *P. citrinum*; lane 13, *A. fumigatus*.

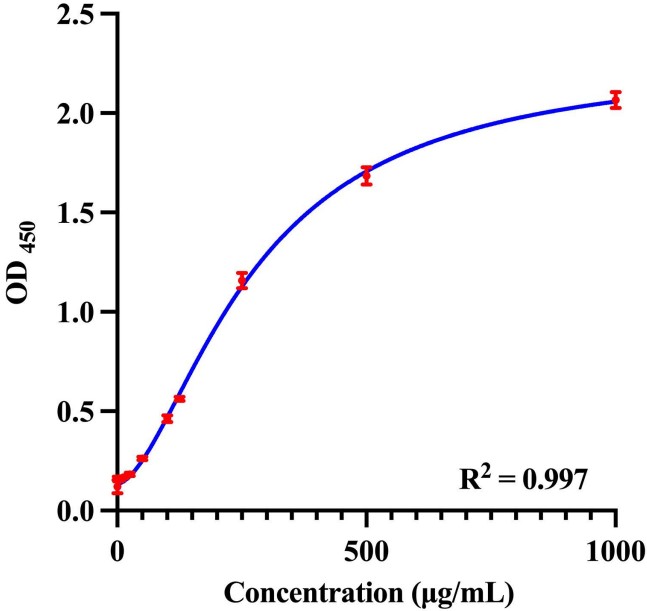

**Fig 4. Standard curve of the biotin-labeled 4D1 sandwich ELISA for the detection of TMCYA.** A standard curve was generated using serial dilutions of pooled serum spiked with TMCYA at concentrations ranging from 0.781 to 1000 µg/mL. Each concentration was assayed in triplicate, and the results are presented as mean ± standard deviation (SD) from three independent experiments. The curve was fitted using a four-parameter logistic (4-PL) regression model and demonstrated excellent reproducibility. Within this concentration range, the coefficient of determination ($R^2$) between the OD values and TMCYA concentrations was 0.997, indicating a highly reliable fit. The detection limit of the assay for *T. marneffei* antigen spiked in pooled serum was approximately 19.398 µg/mL.

minimal deviation between the experimental data points and the fitted curve. Based on the standard curve, the detection limit for *T. marneffei* antigen spiked into pooled serum was determined to be approximately 19.398 µg/mL.

### 3.4 Analytical performance of the assay (LOB, LOD, and LOQ)

To further evaluate the analytical performance of the assay, the LOB, LOD, and LOQ were determined. The analytical LOB was calculated to be 0.168, and the LOD was determined to be 0.175 based on low-concentration samples. The OD value corresponding to the LOD was converted to antigen concentration using the four-parameter logistic (4-PL) standard curve, yielding an LOD of 19.398 µg/mL. The LOQ was defined as the lowest concentration that could be quantified with acceptable precision (CV < 20%). Based on the observed variability of low-concentration samples, the LOQ was estimated to be 25.0 µg/mL. The determination of LOB, LOD, and LOQ is illustrated in Fig G in S1 File.

### 3.5 Specificity of the biotin-labeled 4D1 sandwich ELISA

To assess the specificity of the biotin-labeled 4D1 sandwich ELISA, we tested serial concentrations (0.781-1000 µg/mL) of cytoplasmic antigens from *T. marneffei* (yeast, mold, and conidia) and 9 other medically relevant fungi. As shown in Fig 5, only TMCYA produced a strong and proportional increase in OD$_{450}$ values across the tested concentration range, while *T. marneffei* mold and conidia generated only baseline-level signals similar to the non-*T. marneffei* fungi. Two-way ANOVA demonstrated a highly significant interaction between fungal species and antigen concentration ($P < 0.0001$), confirming that the observed dose-dependent response is unique to TMCYA. These findings indicate that the biotin-labeled 4D1 sandwich ELISA is highly species- and phase-specific, recognizing only the TMCYA without cross-reactivity to other fungal species.

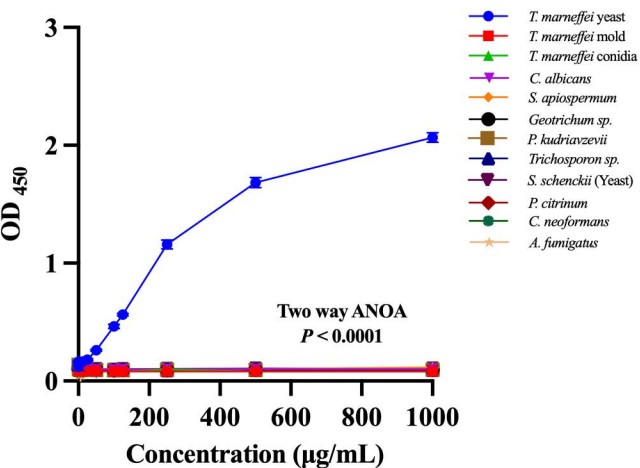

**Fig 5. The specificity of the biotin-labeled 4D1 sandwich ELISA.** Serial concentrations (0.781-1000 μg/mL) of cytoplasmic antigens from *T. marneffei* (yeast, mold, and conidia) and 9 non-*T. marneffei* fungi were evaluated using the biotin-labeled 4D1 sandwich ELISA. Each concentration was assayed in duplicate across three independent experiments, and data are presented as mean±SD. A clear dose-dependent increase in $OD_{450}$ was observed exclusively for TMCYA, whereas *T. marneffei* mold and conidia as well as all other fungal antigens produced only minimal background signals without a detectable response curve. Two-way ANOVA revealed a highly significant interaction between fungal species and antigen concentration ($P<0.0001$), demonstrating that only TMCYA is specifically recognized by the assay.

### 3.6 Detection of serum samples by the biotin-labeled 4D1 sandwich ELISA and diagnostic performance

In total, 460 serum samples were subjected to the biotin-labeled 4D1 sandwich ELISA, including 79 samples from patients with *T. marneffei* infection and 381 samples from the control group without *T. marneffei* infection. The $OD_{450}$ values of the *T. marneffei* infection group and the control group were significantly different (Fig 6A). This result implied that mAb 4D1 was specifically reactive to TMCYA in serum, allowing the biotin-labeled 4D1 sandwich ELISA to distinguish patients with *T. marneffei* infection from controls. The ROC curve was constructed by plotting sensitivity (true positives) versus the corresponding 1-specificity (false positives) at various OD cut-off values (Fig 6B). The area under the ROC curve (AUC) is an important index of accuracy of the curve [57].

In this study, the AUC was 0.983 (95% CI: 0.970-0.993, $P<0.0001$), which indicated that the biotin-labeled 4D1 sandwich ELISA offered excellent detectability of talaromycosis (Fig 6B). The operating point in the upper left corner of the curve indicated by the arrow was chosen as the cut-off with an $OD_{450}$ value of 0.268 (Fig 6A). This cut-off was determined based on the maximum Youden index derived from ROC curve analysis, providing the optimal balance between sensitivity and specificity (Youden index=sensitivity+specificity−1) [58]. The corresponding diagnostic performance was illustrated in Tables 3 and 4. The biotin-labeled 4D1 sandwich ELISA correctly diagnosed *T. marneffei* infection in 70 of 79 (88.61%) cases (95% CI, 79.47%-94.66%), suggesting no significant difference compared to the culture assay ($P>0.05$). The test correctly excluded *T. marneffei* infection in 366 of 381 (96.06%) control individuals (95% CI, 93.59%-97.78%). The accuracy was 94.78% (95% CI, 92.34%-96.63%). The positive likelihood ratio (LR) was 22.51, indicating an association with disease. The negative LR was 0.12, suggesting an association with the absence of disease. Overall, there was a practically perfect agreement between the biotin-labeled 4D1 sandwich ELISA and the culture assay (kappa=0.824, 95% CI: 0.739-0.885, $P<0.001$).

## 4. Discussion

*T. marneffei* is an important opportunistic fungus typically found in patients with secondary immunodeficiencies, especially in those with advanced HIV infection or another cellular immune dysfunction [35,59]. Early and accurate laboratory

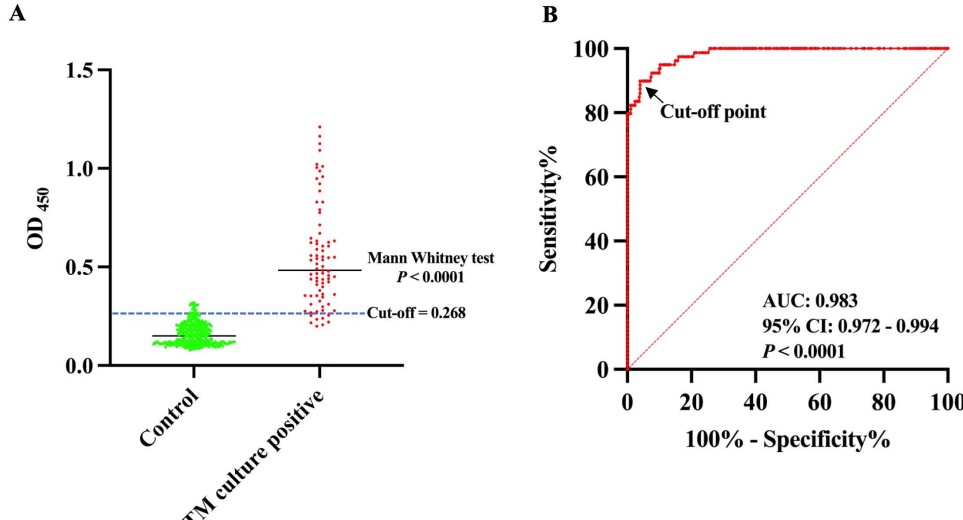

**Fig 6. Diagnostic performance of the biotin-labeled 4D1 sandwich ELISA for detecting TMCYA in serum.** (A) Comparison of $OD_{450}$ values between culture-confirmed *T. marneffei*-positive patients and the control group. Each point represents an individual serum sample. The median $OD_{450}$ in the culture-positive group was significantly higher than that of controls (Mann-Whitney test, *P* < 0.0001). A diagnostic cut-off value of 0.268 (dashed blue line) was determined based on ROC curve analysis. (B) Receiver operating characteristic (ROC) curve of the assay. The area under the curve (AUC) was 0.983 (95% CI: 0.972-0.994, *P* < 0.0001), indicating excellent diagnostic accuracy. The optimal cut-off point was identified at the maximal Youden index.

**Table 3. Diagnostic performance of the biotin-labeled 4D1 sandwich ELISA compared with culture (reference standard).**

| Culture status | ELISA positive | ELISA negative | Total |
|---|---|---|---|
| Positive | 70 | 9 | 79 |
| Negative | 15 | 366 | 381 |
| Total | 85 | 375 | 460 |

**Note:** TP, true positive; FP, false positive; FN, false negative; TN, true negative. Culture was used as the reference standard. The cut-off value for ELISA positivity was determined by ROC analysis.

**Table 4. Diagnostic performance of the biotin-labeled 4D1 sandwich ELISA when testing serum samples from individuals with or without *T. marneffei* infection.**

| Diagnostic performance criteria | Biotin-labeled 4D1 sandwich ELISA | |
|---|---|---|
| | Value | 95%CI |
| Sensitivity | 88.61% | 79.47%-94.66% |
| Specificity | 96.06% | 93.59%-97.78% |
| Positive Likelihood Ratio (LR+)* | 22.51 | 13.62-37.19 |
| Negative Likelihood Ratio (LR−)* | 0.12 | 0.06-0.22 |
| Accuracy | 94.78% | 92.34%-96.63% |
| Kappa# | 0.822 | 0.753-0.891 |

* The likelihood ratios were interpreted as follows: LR+ > 10 or LR− < 0.1 were considered strong diagnostic evidence; LR+ between 5–10 or LR− between 0.1-0.2 indicated moderate evidence; and values approaching 1 suggested limited diagnostic utility.

# The Kappa value can be interpreted: Strength of agreement: < 0.20, Poor; 0.21-0.40, Fair; 0.41-0.60, Moderate; 0.61-0.80, Good; 0.81-1.00, Very good.

diagnosis is essential to improve prognosis and guide prompt antifungal therapy. Currently, the definitive diagnosis of *T. marneffei* still relies on fungal culture and morphological characterization, but this process is time-consuming and often leads to delayed treatment and increased morbidity and mortality [59].

In this study, we developed and systematically evaluated a biotin-labeled mAb 4D1-based sandwich ELISA for the detection of the cytoplasmic yeast antigen of *T. marneffei* in serum, building upon the previously validated specificity of mAb 4D1 [29,33,35,60]. Compared with conventional culture and previously reported immunoassays, this method demonstrated good diagnostic performance, with a high area under the curve (AUC = 0.983), high sensitivity (88.61%) and specificity (96.06%), and good concordance with culture results (κ = 0.822). The high positive likelihood ratio (22.51) and low negative likelihood ratio (0.12) further support the assay's discriminatory ability. Collectively, these findings indicate that the developed assay has considerable potential for clinical application. Notably, this method enables rapid quantitative detection, and the four-parameter logistic regression model exhibited strong goodness-of-fit ($R^2$ = 0.997) across a broad range of antigen concentrations (0.781-1000 µg/mL). This quantitative performance suggests potential utility for monitoring antigen dynamics; however, this application was not evaluated in the present study and requires further validation.

Compared with other diagnostic methods, the biotin-labeled 4D1 sandwich ELISA combines high sensitivity with practical applicability. Although molecular assays, such as metagenomic next-generation sequencing (mNGS) and multiplex PCR can detect *T. marneffei*, these methods are costly, technically complex, and may have limited specificity [14,24,25]. In immunological methods, commonly used targets such as galactomannan (GM) or Mp1p have shown variable performance, with reported sensitivities and specificities generally ranging from approximately 70% to over 90% depending on study design and patient populations [16,26,27,61]. In comparison, the biotin-labeled 4D1 sandwich ELISA developed in this study achieved a sensitivity of 88.61% and a specificity of 96.06%, indicating comparable or potentially improved diagnostic performance. Previous studies have reported that antibody-based detection of *T. marneffei*, including Mp1p-related assays, may exhibit cross-reactivity with other pathogens [26,62]. In contrast, mAb 4D1 has been shown to recognize high-molecular-weight mannoproteins of *T. marneffei* (50–180 kDa), providing a basis for the development of highly specific diagnostic reagents [27,29,30,34,35]. In addition, the present assay provides results within a few hours, whereas conventional culture methods require 7–14 days, highlighting its advantage in turnaround time.

The biotin-streptavidin amplification system further enhanced detection sensitivity, enabling efficient signal amplification even at low antigen concentrations [63]. Notably, the use of mAb 4D1 as both the capture antibody and its biotin-conjugated counterpart as the detection antibody establishes a dual-recognition format, which enhances assay specificity while minimizing nonspecific binding. In this study, the assay exhibited no cross-reactivity with the tested fungal antigens, supporting its specificity for *T. marneffei* and highlighting its potential for clinical application. Importantly, the biotin-labeled 4D1 sandwich ELISA provides a rapid and accurate diagnostic approach for immunocompromised patients, with results obtainable within hours, compared with conventional culture methods that require 7–14 days. Compared with conventional culture, the assay can be performed using standard laboratory equipment without the need for prolonged incubation or advanced molecular platforms, suggesting potential advantages in resource-limited and endemic settings.

Despite the encouraging results of this study, several limitations remain. First, the lowest detection limit was 19.398 µg/mL, which may be attributable to the use of the same monoclonal antibody for both capture and detection, potential interference from serum components, and the complexity of the sample matrix. Compared with commercially available cryptococcal antigen assays, which have demonstrated sensitivities and specificities exceeding 99% in some clinical settings [64,65], the analytical sensitivity of the present assay remains suboptimal and warrants further optimization. Second, some of the healthy controls were from endemic areas, and the possibility of undetected infection could not be completely excluded due to lack of culture and follow-up information. Although detailed clinical characteristics, including age, sex, and CD4+ T-cell counts, are summarized in the supplementary tables, incomplete clinical data remained for a subset of patients. In particular, HIV infection status could not be accessed due to institutional ethical restrictions, and CD4+ T-cell counts were unavailable for all cases. Therefore, the impact of host immune status on diagnostic performance

could not be fully assessed. Furthermore, prolonged storage of some archived serum samples may have affected antigen stability and assay performance. As this was a single-center study, a future multicenter validation in endemic regions, including Thailand, Vietnam, and southern China, is required to assess broader applicability. Future studies should also evaluate the cost-effectiveness of this assay and explore its adaptation into a lateral flow assay (LFA) or point-of-care testing (POCT) formats, as well as its potential integration with complementary biomarkers such as β-D-glucan or galactomannan.

Overall, the biotin-labeled 4D1 sandwich ELISA integrates simplicity and good diagnostic performance, supporting its potential utility for rapid screening and clinical surveillance of *T. marneffei* infection, particularly in resource-limited settings, as it can be performed using standard laboratory equipment with a short turnaround time and without the need for prolonged culture or advanced molecular platforms.

## 5. Conclusion

This study evaluated the diagnostic performance of a biotin-labeled 4D1-based sandwich ELISA for the detection of *T. marneffei* yeast antigen in serum. The assay demonstrated good diagnostic performance, with high sensitivity, specificity, and good agreement with culture-based diagnosis, while substantially reducing the time required for pathogen detection. These findings suggest that the assay offers a simple and quantitative approach for the diagnosis of talaromycosis and may be useful in endemic settings. Further studies are warranted to evaluate its cost-effectiveness and its potential utility for disease monitoring.

## Supporting information

**S1 File. Table A.** Detailed clinical characteristics of patients with culture-confirmed *T. marneffei* infection. **Table B.** Detailed microbiological findings and clinical characteristics of *T. marneffei*-negative patients with other microbial infections. **Table C.** Optimization of capture and detection antibody conditions for the sandwich ELISA. **Table D.** Optimization of streptavidin-HRP dilution in the sandwich ELISA. **Table E.** Optimization of serum dilution for antigen detection in the sandwich ELISA. **Fig A.** Biotinylation strategy showing NHS-ester activation and subsequent conjugation of avidin/streptavidin to primary amine-containing proteins. **Fig B.** Purification of mAb 4D1. **Fig C.** Controls validating the specificity of mAb 4D1 and its biotin-labeled conjugate in Western blot detection of TMCYA. **Fig D.** Optimization of capture and detection antibody conditions for the sandwich ELISA. **Fig E**. Optimization of streptavidin-HRP dilution in the sandwich ELISA. **Fig F.** Optimization of serum dilution for antigen detection in the sandwich ELISA. **Fig G.** Determination of analytical performance parameters (LOB, LOD, and LOQ) of the biotin-labeled 4D1 sandwich ELISA.
(DOCX)

## Author contributions

**Conceptualization:** Huamei Wei, Artid Amsri, Patcharin Thammasit, Kritsada Pruksaphon, Sirida Youngchim.

**Data curation:** Huamei Wei, Artid Amsri, Kritsada Pruksaphon, Yuanji Teng, Changke Pu, Yuefeng Huang.

**Formal analysis:** Artid Amsri, Patcharin Thammasit, Kritsada Pruksaphon, Sirida Youngchim.

**Funding acquisition:** Huamei Wei, Sirida Youngchim.

**Investigation:** Huamei Wei, Kritsada Pruksaphon, Joshua D Nosanchuk.

**Methodology:** Huamei Wei, Artid Amsri, Kritsada Pruksaphon.

**Project administration:** Patcharin Thammasit, Sirida Youngchim.

**Resources:** Kritsada Pruksaphon, Yuanji Teng, Changke Pu, Yuefeng Huang.

**Supervision:** Artid Amsri, Patcharin Thammasit, Joshua D Nosanchuk, Sirida Youngchim.

**Visualization:** Huamei Wei, Artid Amsri.

**Writing – original draft:** Huamei Wei, Artid Amsri, Patcharin Thammasit.

**Writing – review & editing:** Huamei Wei, Artid Amsri, Patcharin Thammasit, Joshua D Nosanchuk, Sirida Youngchim.

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
