## [Decision Letter · Decision Letter 0]

19 Mar 2026

PNTD-D-26-00326

Diagnostic performance of a biotin-labeled 4D1 sandwich ELISA for serum antigen detection in talaromycosis

Dear Dr. Youngchim,

Thank you for submitting your manuscript to PLOS Neglected Tropical Diseases. After careful consideration, we feel that it has merit but does not fully meet PLOS Neglected Tropical Diseases's publication criteria as it currently stands. Therefore, we invite you to submit a revised version of the manuscript that addresses the points raised during the review process.

Please submit your revised manuscript within by May 18 2026 11:59PM. If you will need more time than this to complete your revisions, please reply to this message or contact the journal office at plosntds@plos.org. Please include the following items when submitting your revised manuscript:

We look forward to receiving your revised manuscript.

Kind regards,

Angel Gonzalez, Ph.D.

Academic Editor

Marcio Rodrigues

Section Editor

Shaden Kamhawi

co-Editor-in-Chief

Paul Brindley

co-Editor-in-Chief

**Journal Requirements:**

At this stage, the following Authors/Authors require contributions: Huamei Wei, Artid Amsri, Patcharin Thammasit, Kritsada Pruksaphon, Yuanji Teng, Changke Pu, Yuefeng Huang, Joshua D Nosanchuk, and Sirida Youngchim. Please ensure that the full contributions of each author are acknowledged in the "Add/Edit/Remove Authors" section of our submission form.

- ® on page: 11.

- TM on page: 11.

3) Tables should not be uploaded as individual files. Please remove these files and include the Tables in your manuscript file as editable, cell-based objects. For more information about how to format tables, see our guidelines:

https://journals.plos.org/plosntds/s/tables

4) We notice that your supplementary Figures, and Tables are included in the manuscript file. Please remove them and upload them with the file type 'Supporting Information'. Please ensure that each Supporting Information file has a legend listed in the manuscript after the references list.

**Reviewers' Comments:**

Reviewer's Responses to Questions

**Key Review Criteria Required for Acceptance?**

**Methods:**

-Are the objectives of the study clearly articulated with a clear testable hypothesis stated?

-Is the study design appropriate to address the stated objectives?

-Is the population clearly described and appropriate for the hypothesis being tested?

-Is the sample size sufficient to ensure adequate power to address the hypothesis being tested?

-Were correct statistical analysis used to support conclusions?

-Are there concerns about ethical or regulatory requirements being met?

Reviewer #1: (No Response)

Reviewer #2: The authors describe a newly developed ELISA for the detection of T. marnaffei cytoplasmic antigen detection in serum.

Overall the study design is suitable to address stated objectives.

Nevertheless there are some points I want to address:

- Line 200: please include expexted values

- starting Line 222: Plesse describe how concentration of antigens was determined

- Secition 2.6: Should describe the development of the assay but only descibes the final setup. It is lacking information on how ideal coating concentration, sample and conjugate dilution were idetified.

- Line 327: are these samples part of the sample panels described in 2.1 and coverd by the respective IRB vote?

- Section 2.7: I recommend splitting up in Standard Curve Costruction, sample testing and specificity testing to improve readablity

- line 329: how were serial dilutions performed? Which stating-dilution or concentration was used?

- Methodology for determination of Limit of Detection is missing

- 4-PL is not described in methods

Reviewer #3: The methodology is very clear, and the hypothesis and objectives are well defined; I think we managed to cover the proposed topic well.

**Results:**

-Does the analysis presented match the analysis plan?

-Are the results clearly and completely presented?

-Are the figures (Tables, Images) of sufficient quality for clarity?

Reviewer #1: (No Response)

Reviewer #2: - Showing results of the development process (according to 2.6) would increase the value of the data, by demonstrating that optimal conditions were determined and chosen.

- Figure 3: Fungal antigens seem to have quite different concentrations. This may bias specificity testing. Was crossreactivity expected with the different fungal antigens? Is there any literature on this?

- it is unclear how the LoD has been determined. For a qanttitative assay Limit of Blank an Limit of Quantification are missing.

- Line 454: Why was this cut-off chosen?

- line 525: There is no data on bacterial or viral antigens; In my opinion this is not necessary to test, but then it should not be stated

- line 552: Please include data on costs to assess cost-effectiveness

Reviewer #3: The results were consistent with the proposal, although some data were presented in a confusing manner; more detailed comments follow below:

Supplementary Figure 3 (A, B, and C) is confusing and difficult to understand what is being shown. Perhaps the quality could be improved, and the bands and sizes of interest could be indicated.

The expected sizes in the gel in Figure 3 could also be better indicated.

The table 3 is confusing; I suggest inverting it, showing how many of the positive and negative (total) included in the study were analyzed, and then showing the results of the biotin-labeled 4D1 sandwich ELISA.

**Conclusions:**

-Are the conclusions supported by the data presented?

-Are the limitations of analysis clearly described?

-Do the authors discuss how these data can be helpful to advance our understanding of the topic under study?

-Is public health relevance addressed?

Reviewer #1: (No Response)

Reviewer #2: This section should be revised and formulated more precise.

Line 556: the manuscript describes selected performance data of the assay, but not its development

Line 558: "the term "strong sensitivity" contradicts line 536/537

Evidence to support cost-effectiveness and usability in treatment monitoring are missing. Finally it is unclear to me why this assay is especially suitable in ressource-limited and endemic settings.

The relevance for public health of such an assay has in general been sufficiently addressed.

Reviewer #3: Yes, the topic is relevant and I believe it opens doors to new diagnostic possibilities and fills an important gap. The limitations were well explored in the discussion.

**Editorial and Data Presentation Modifications?**

Reviewer #1: (No Response)

Reviewer #2: Figure 1: This figure is used to descibe the diffenrent cohorts of patients from which samples for this study have been derived. A Flow-diagram seems unsuitable for this pupose. I rather recommend a table.

Concerning expected sensitivity and specificity as well as the comparison of the newly developed assay to other assays (starting line 511), there are only reference given. For the reader it would be very beneficial to state these numbers and include a direct comparison in the manuscript. Additionally this would underline advantages of the new assay compared to others.

The abbrevation "TM CYA" is not used consitently (alternative: TMCYA).

Figure 2: This figure does not show the development (which is a process) of an assay but the principle of a sandwich ELISA or the final setup of this specific sandwich ELISA.

Reviewer #3: Suggestions: "292 The second membrane was incubated with biotin-labeled 4D1 (20:1, diluted 1:1000 in 1% (w/v) BSA prepared in 10 mM Tris-HCl, pH 7.4) for 2 hours and subsequently incubated with streptavidin-HRP diluted 1:6000 in 1% (w/v) BSA prepared in 10 mM Tris-HCl (pH 7.4) for 1 hour." Explain how you arrived at this dilution of biotin-labeled 4D1.

524 "Moreover, the assay exhibited no cross-reactivity wit common fungal, bacterial, or viral antigens, demonstrating its strong potential for clinical application"

No bacterial or viral data was shown here; therefore, add a reference that shows this information.

**Summary and General Comments:**

Reviewer #1: 1. Introduction: The statement, "Moreover, many studies .... detection of T. marneffei [36, 37]", requires clarification. The authors are encouraged to ensure the references 36 and 37 support this claim regarding T. marneffei.

2. An informed consent statement was not provided in the methods section. Please clarify whether informed consent was obtained.

3. Methods - serial dilution: The rationale for using a 1:10 dilution should be clarified. The authors are encouraged to provide a brief explanation or cite an appropriate reference supporting the use of this dilution.

Reviewer #2: This manuscript describes the evaluation of a T. marnaffei ELISA for detection of cytoplasmic antigen. The overall design appears suitable for the stated opjectives.

In my opinion especially the clinical performance was very well adressed by using an excellently characterized sample panel. The assay resulting from this work can very well support diagnosis of Talaromycosis.

However, substantial methodological details required for reproducibility and proper evaluation of assay performance are currently missing or insufficiently described. Additionally the desired outcome should be stated more clearly and followed more stringently (Performance focus or development focus).

In its present form, I would therefore consider this manuscript suitable only after major revision.

Reviewer #3: (No Response)

PLOS authors have the option to publish the peer review history of their article (what does this mean?). If published, this will include your full peer review and any attached files.

Reviewer #1: No

Reviewer #2: No

Reviewer #3: No

**Figure resubmission:** While revising your submission, we strongly recommend that you use PLOS’s NAAS tool (https://ngplosjournals.pagemajik.ai/artanalysis) to test your figure files. NAAS can convert your figure files to the TIFF file type and meet basic requirements (such as print size, resolution), or provide you with a report on issues that do not meet our requirements and that NAAS cannot fix.
---

## [Editor Report · Decision Letter 1]

28 Apr 2026

Dear Dr. Youngchim,

We are pleased to inform you that your manuscript 'Diagnostic performance of a biotin-labeled 4D1 sandwich ELISA for serum antigen detection in talaromycosis' has been provisionally accepted for publication in PLOS Neglected Tropical Diseases.

Best regards,

Angel Gonzalez, Ph.D.

Academic Editor

Marcio Rodrigues

Section Editor

Shaden Kamhawi

co-Editor-in-Chief

Paul Brindley

co-Editor-in-Chief

---

## [Editor Report · Acceptance letter]

Dear Dr Youngchim,

We are delighted to inform you that your manuscript, "Diagnostic performance of a biotin-labeled 4D1 sandwich ELISA for serum antigen detection in talaromycosis," has been formally accepted for publication in PLOS Neglected Tropical Diseases.

Best regards,

Shaden Kamhawi

co-Editor-in-Chief

Paul Brindley

co-Editor-in-Chief
